# COMPOSEANYTHING: COMPOSITE OBJECT PRIORS FOR TEXT-TO-IMAGE GENERATION

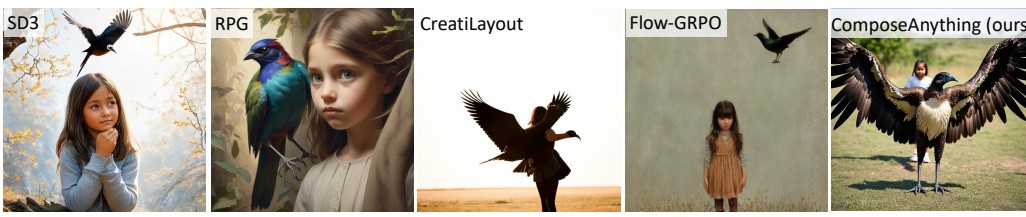

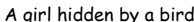

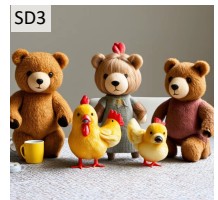 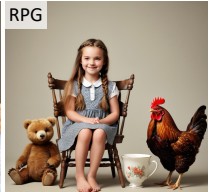 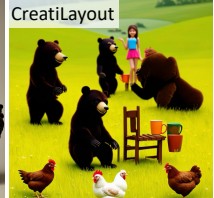 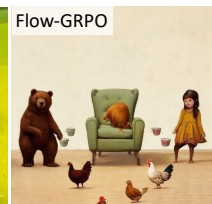 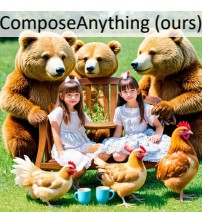

Figure 1: *ComposeAnything* enables text-to-image generation for complex compositions involving surreal spatial relationships and high object counts. Unlike layout-conditioned (*e.g.*, RPG (Yang et al., 2024) and CreatiLayout (Zhang et al., 2024a)), or reinforcement learning methods (*e.g.*, Flow-GRPO (Liu et al., 2025), it achieves both high visual quality and strong faithfulness to text.

## ABSTRACT

Generating images from text with complex object arrangements remains a major challenge for current text-to-image (T2I) models. Existing training-based solutions, such as layout-conditioned models or reinforcement learning methods, improve compositional accuracy but often distort realism, leading to floating objects, broken physics, and degraded image quality. In this work, we introduce ComposeAnything, an inference-only framework that enhances compositional generation without re-training. Our key idea is to replace stochastic noise initialization with *composite object priors*— interpretable structured composite of objects, created using 2.5D layouts generated from large language models and pretrained image generators. We further propose prior-guided diffusion, which integrates these priors into the denoising process to enforce compositional correctness while preserving visual fidelity. This training-free strategy enables seamless generation of compositional objects and coherent backgrounds, while allowing refinement of inaccurate priors. ComposeAnything consistently outperforms state-of-the-art inference-only methods on T2I-CompBench and NSR-1K benchmarks, especially for prompts with complex spatial relations, high object counts, and surreal scenes. Human evaluations confirm that our method generates images that are not only compositionally faithful but also visually coherent.

## 1 INTRODUCTION

Text-to-image (T2I) models, particularly diffusion-based ones such as SDXL (Podell et al., 2023), SD3 (Esser et al., 2024) and Flux (Black Forest Labs, 2024), have achieved remarkable success in generating individual concepts with high fidelity. However, they struggle with complex object compositions (Huang et al., 2023), especially novel arrangements that deviate from their training

distribution, often resulting in unnatural mixing of objects, incorrect 2D/3D spatial positioning, and inaccurate object counts, as shown in Figure 1.

To improve compositional generation, prior work has explored layout control and reinforcement learning (RL). Layout-based methods use 2D cues (e.g., boxes or blobs) often derived from LLMs (OpenAI, 2025) to steer generation (Feng et al., 2023b; Li et al., 2023a; Nie et al., 2024). Training-based variants adapt pretrained T2I models with layout-conditioning modules (Zhang et al., 2024a; Li et al., 2023a; Zhang et al., 2023a; Wang et al., 2024a), offering stronger spatial control but incurring heavy training cost and quality degradations under rigid constraints (Zhang et al., 2024b) (e.g., CreatiLayout (Zhang et al., 2024a) in Figure 1). Inference-only variants guide denoising via attention/latent manipulation or region-wise denoising (Yang et al., 2024; Chefer et al., 2023; Dahary et al., 2024), preserving quality better but providing weaker control for unusual layouts, higher object counts, and 3D relations because they rely on *coarse 2D* signals without appearance priors.

RL-based methods such as DDPO (Black et al., 2024), DPOK (Fan et al., 2023) and Flow-GRPO (Liu et al., 2025) optimize explicit compositional rewards (e.g., counts and spatial relations) to enforce alignment. While this improves compositional scores, the reward-driven optimization tends to overexploit the imperfect reward signal, yielding floating objects, faded backgrounds, and broken physical realism — trading fidelity for composition as shown in Figure 1 and 7. Moreover, rigid box-conditioned training overfits to layout constraints, compromising image coherence.

We propose *ComposeAnything* to address these limitations with a purely *inference-time* solution that balances accurate composition and visual realism. The key idea is to replace stochastic noise in pretrained diffusion models with *composite object priors*: structured object-level priors created from text using LLM reasoning and off-the-shelf image generators. These priors carry appearance, count/size, and coarse 2.5D placement with depth cues, going beyond box-only cues. We then introduce *prior-guided diffusion*, which integrates the priors in the early stage of denoising. It combines *object-prior reinforcement* and *spatially controlled denoising*. The former preserves foreground priors in early steps while allowing the model to synthesize coherent backgrounds; the latter strengthens the spatial arrangement of the composite prior via mask-guided attention in early diffusion steps where global structure is determined. After these initial steps, we revert to standard diffusion to refine detail and realism. *ComposeAnything* outperforms state-of-the-art inference-only methods on T2I-CompBench (Huang et al., 2023) and NSR-1K (Feng et al., 2023b) under automatic metrics, and achieves significant improvement over human evaluations over all the baselines. Ablations confirm the contributions of composite object priors and prior-guided diffusion.

**Contributions.** (i) A training-free interpretable framework that replaces random noise with *composite object priors* carrying appearance and coarse 2.5D structure derived from text via LLMs and pretrained generators. (ii) *Prior-guided diffusion* that integrates these priors via object-prior reinforcement and spatially controlled denoising in early steps, balancing compositional fidelity and image quality. (iii) State-of-the-art quality–compositionality trade-off on challenging benchmarks, particularly for surreal spatial relations, high-object-count, and generally complex prompts; code to be released.

## 2 RELATED WORKS

**Compositional generation.** Compositional T2I aims to produce images that faithfully reflect complex textual descriptions (Huang et al., 2023; Zhang et al., 2024b; Jamwal & S., 2024; Li et al., 2024a; Wang et al., 2024a; Feng et al., 2023b; Yang et al., 2024; Wang et al., 2024b; Couairon et al., 2023; Lian et al., 2024). While modern diffusion models (Podell et al., 2023; Esser et al., 2024; Black Forest Labs, 2024) are strong generators, they struggle with novel multi-object arrangements, spatial relations, and counting.

**Training-based layout control.** A common direction is to inject explicit spatial conditioning during training. Methods fine-tune pretrained backbones (Podell et al., 2023; Esser et al., 2024; Black Forest Labs, 2024; Chen et al., 2023; 2024a; Pernias et al., 2023) or add adapters with grounding/alignment objectives (Wang et al., 2024b; Jiang et al., 2024; Hu et al., 2024a). Layout-controlled variants train conditioning modules for boxes, masks, or keypoints (Li et al., 2023a; Zhang et al., 2023a; Wang et al., 2024a; Feng et al., 2024; Yang et al., 2023; Zhang et al., 2024a; Lin et al., 2025; Zhao et al., 2023; Zheng et al., 2023; Gani et al., 2024; Mou et al., 2024; Li et al., 2024b). These approaches

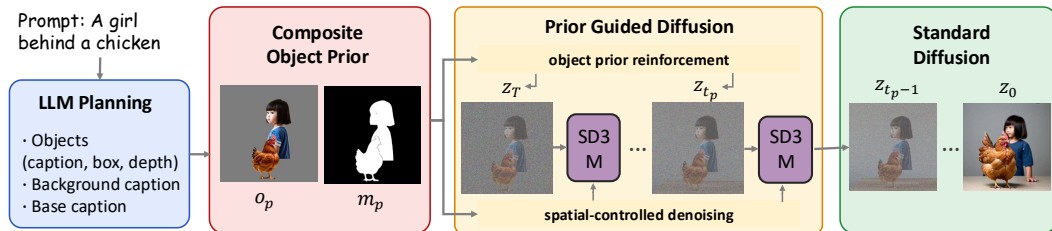

Figure 2: The *ComposeAnything* framework, which enhances text-to-image diffusion models with layouts and composite object priors for complex compositional generation.

can enforce geometry but require substantial training and often degrade coherence and realism under hard constraints (Zhang et al., 2024b).

**RL-based compositional control.** Reinforcement learning optimizes explicit rewards for counts and spatial relations (e.g., Flow-GRPO (Liu et al., 2025), DDPO (Black et al., 2024), DPOK (Fan et al., 2023)). Such rewards improve compositional scores but can induce distribution shift: models may exploit the reward at the expense of realism (floating objects, faded backgrounds, broken physics), even with KL regularization. This trades fidelity for composition, complementary to hard box-conditioned training.

**Training-free (inference-only) control.** Another line manipulates pretrained models at inference time, avoiding retraining. Attention- and latent-based methods edit text embeddings or cross-attention to steer local content (Chefer et al., 2023; Feng et al., 2023a; Meral et al., 2024; Trusca et al., 2024; Li et al., 2023b; Rassin et al., 2023; Liu et al., 2022; Agarwal et al., 2023; Gong et al., 2024) and perform region-wise denoising (Yang et al., 2024; Li et al., 2024c). Layout-driven, training-free techniques use LLM-derived 2D layouts (boxes/blobs) to modulate attention or emphasize regions (OpenAI, 2025; Feng et al., 2023b; Li et al., 2023a; Nie et al., 2024; Zhang et al., 2024b; Xie et al., 2023; Dahary et al., 2024; Kim et al., 2023; Ma et al., 2024; Couairon et al., 2023; Chen et al., 2024b; Jamwal & S., 2024; Phung et al., 2024). These preserve base-model quality better than training-based approaches, but control is weaker and brittle for unusual layouts, high object counts, and 3D relations because guidance is limited to *coarse 2D* signals without appearance priors.

**Inference-time noise search & initialization.** A complementary thread exploits the sensitivity of diffusion to the initial condition—either by searching/optimizing seeds and trajectories (Ma et al., 2025; Guo et al., 2024). These approaches can boost success rates but are compute-intensive and brittle for out-of-distribution, highly compositional prompts. Prior work in image editing leverages noisy initialization/inversion for image to image translation (Meng et al., 2022; Avrahami et al., 2022; Mao et al., 2023).

We instead *generate* the initial condition as *composite object priors*—coarse RGB composites encoding appearance and coarse 2.5D layout, and integrate them with *prior-guided diffusion*. Unlike training-based or RL methods, our approach is *inference-only* and avoids reward-driven distribution shift; unlike prior inference-only methods, it goes beyond attention tweaks and 2D boxes by injecting appearance-aware priors that deliver stronger compositional control while preserving realism.

## 3 THE PROPOSED METHOD

As illustrated in Figure 2, our *ComposeAnything* framework consists of three key components for compositional text-to-image generation: 1) LLM Planning (Section 3.1): We employ LLMs to transform the input prompt into a structured 2.5D semantic layout, including object captions, bounding boxes and relative depths; 2) Composite Object Prior (Section 3.2): Based on the layout, we generate a coarse composite image that serves as a strong semantic and spatial prior for guiding image synthesis; and 3) Prior Guided Diffusion (Section 3.3): We iteratively initialize noises with the object prior and apply spatially-controlled self-attention to preserve structure in early denoising steps.

### 3.1 LLM PLANNING

Recent advancements in LLMs have demonstrated their effectiveness in generating high-quality scene layouts from textual descriptions (Feng et al., 2023b; Yang et al., 2024; Hu et al., 2024b).

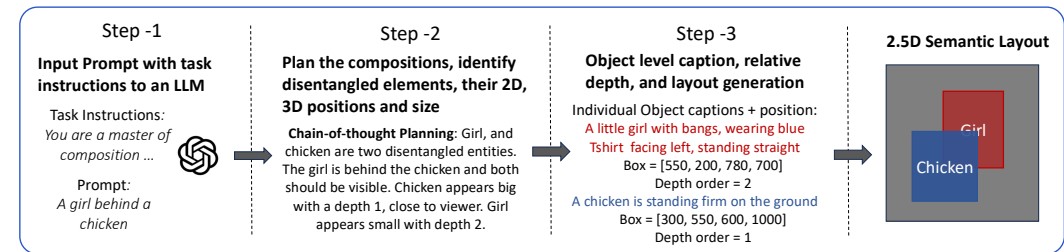

Figure 3: Chain-of-thought LLM planning for generating 2.5D semantic layouts from text.

Hence, we harness GPT-4.1 (OpenAI, 2025) to produce a structured 2.5D semantic layout from the original text. The layout includes the following elements: Object captions $\{y_{o_i}\}_{i=1}^K$ that describe size, orientation and appearance for each identified object; Bounding boxes $\{box_i\}_{i=1}^K$ that specify 2D spatial configuration for each object; Depth values $\{depth_i\}_{i=1}^K$ that reflect relative depth orders for each object to support 3D-aware composition; Background caption $y_{bg}$ describing the background scene; and Compositional caption $y_{base}$ which is a concise summary of the entire image. This process involves several key steps for chain-of-thought reasoning, as illustrated in Figure 3. More details are provided in Appendix E.

## 3.2 COMPOSITE OBJECT PRIOR

**2.5D position-aware composite image generation.** Given the isolated object captions from LLM, we first generate individual objects using Stable Diffusion-3 Medium (SD3-M) (Esser et al., 2024). Next, we use a referring expression segmentation model Hyperseg (Wei et al., 2024) to extract objects $\{o_i\}_{i=1}^K$ along with their segmentation masks $\{m_i\}_{i=1}^K$. Each object and its corresponding mask are resized to fit within the designated bounding box generated from the LLM according to a scaling factor $scale_i$. Objects are then composited in a depth-aware order, where objects with smaller depth values are placed above those with larger depth values, thereby establishing occlusion-correct layering in the final scene. This process is formulated as follows: $o_i', m_i' = \text{Resize}(o_i, m_i, scale_i)$; then $o_p, m_p = \text{Compose}(\{o_i'\}, \{m_i'\}, \{box_i\}, \{depth_i\})$. Finally, all objects are composited on a $N \times N$ sized canvas, denoted as $o_p$. Its corresponding composited mask is denoted as $m_p$. Figure 2 shows an example of the composite image and mask. The composition of all objects forms the foreground, and the rest is considered the background.

**Initializing object prior for diffusion-based models.** Our work builds upon existing T2I diffusion models, aiming to enhance its ability to generate images with complex object compositions. Our method is compatible with both denoising diffusion probabilistic models like SDXL (Podell et al., 2023) and recent flow-matching based models like SD3-M (Esser et al., 2024).

The core idea of diffusion models is to learn a generative process by simulating and then reversing a gradual noising procedure. Given an image $x_0$ from the real data distribution $p(x)$, the forward process transforms $x_0$ into $x_T \sim \mathcal{N}(0, I)$ through a predefined noise schedule:

$$x_t = \alpha(t)x_0 + \sigma(t)z, \; z \sim \mathcal{N}(0, I), \tag{1}$$

where $t \in [0, T]$ indexes the diffusion timestep. A denoising network $\epsilon(\theta)$ is trained to predict the added noise at each step in the forward process. During inference, image generation starts from pure Gaussian noise $x_T$ and denoises it back to $x_0$ via the reverse process, which is an ordinary differential model (ODE) on time $t \in [T, 0]$ guided by the noise prediction network $\epsilon(\theta)$: $x_{t-\Delta t} \leftarrow x_t - \epsilon(\theta)(x_t, t) \Delta t$.

Our method is inspired by the fact that the reverse ODE can be solved from any $t \in (0, T)$ (Meng et al., 2022). Instead of starting from pure Gaussian noise at $t = T$, we initialize the process with a noisy object prior at an intermediate timestep $t_p < T$, providing a stronger starting point for generation. Specifically, we follow latent diffusion models (Rombach et al., 2022) where the denoising is applied on the latent space. We use the above composite image $o_p$ to generate an initial noise in the latent space. The image $o_p$ is first encoded through a Variational Autoencoder (VAE) to get the prior latent. Then, we apply the forward process from Eq. (1) at a high noise timestep $t_p$ to obtain the latent object prior, $z^{o_p} = \text{VAE}(o_p)$, and its noised version $\hat{z}_{t_p}^{o_p} = \alpha(t_p)z^{o_p} + \sigma(t_p)z$, with

$z \sim \mathcal{N}(0, I)$. Since the background in $o_p$ is empty, we avoid conditioning the generation process on the uninformative background region of the latent $\hat{z}_{t_p}^{o_p}$. To achieve this, we use the mask $m_p$ to reinitialize the background with pure Gaussian noise: $z_{t_p}^{o_p} = \hat{z}_{t_p}^{o_p} \odot m_p + z_{bg} \odot (1 - m_p)$. where $z_{bg} \sim \mathcal{N}(0, I)$ and $\odot$ indicates element multiplication. This ensures that only the object regions are guided by a prior, while the background remains free to be generated based on the caption. The reverse process still starts from $t = T$, but uses the composite object prior $z_{t_p}^{o_p}$ as initialization.

### 3.3 PRIOR-GUIDED DIFFUSION

We propose two mechanisms to incorporate the guidance from the composite object prior in the denoising process. Figure 4 illustrates the prior-guided diffusion method.

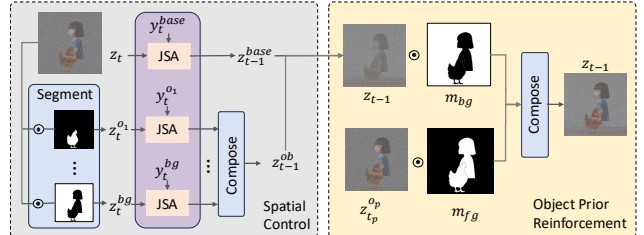

**Object prior reinforcement.** To prevent excessive corruption of the foreground object prior, we initialized the foreground with noise at time $t_p$, while the background is still initialized with pure Gaussian noise at $T$. However, during denoising from $t =$

Figure 4: Overview of prior-guided diffusion. Spatial-controlled denoising is applied for each aligned text and region pair to strengthen spatial control. We further re-inject the object prior $z_{t_p}^{o_p}$ into predicted $z_{t-1}$ to reinforce the prior.

$T$, this mismatch in noise levels leads to inaccurate noise predictions for the foreground region, which potentially distort its semantics and structure. To address this, we propose a novel foreground prior reinforcement algorithm. During the denoising steps from $T$ to $t_p$, we repeatedly restore the original object prior in the foreground regions to protect them from degradation. Specifically, we overwrite the foreground region in the current latent $z_{t-1}$ with the initial object prior, while retaining the denoised background: $z_{t-1} \leftarrow z_{t_p}^{o_p} \odot m_p + z_{t-1} \odot (1 - m_p)$.

This iterative replacement ensures that the semantic integrity and spatial structure of the object prior are preserved throughout the early diffusion steps. At the same time, the background is progressively refined in the presence of a fixed foreground, allowing for coherent integration between the two.

Once the latent reaches time $t_p$, both foreground and background are aligned in terms of noise level and the global structure becomes stable. From this point onward, denoising proceeds without any additional intervention, allowing for natural refinement and generative flexibility. Notably, decreasing $t_p$ strengthens the object prior while reducing generative flexibility.

**Spatial-controlled denoising.** To further enhance object-level spatial control in T2I generation, we propose a spatial-controlled attention mechanism that explicitly strengthens the alignment between between specific image regions and their corresponding region textual descriptions.

Our method builds on Multi-Modal Diffusion Transformers, a dual-stream architecture used in Stable Diffusion 3 (Esser et al., 2024), which processes text and image modalities in parallel. In addition to the base prompt embeddings $y^{base}$, we introduce a set of $K$ object prompt embeddings $\{y^{o_i}\}_{i=1}^{K}$ and one background prompt embedding $y^{bg}$. These are independently processed by the text stream, while the image stream receives only the latent image embeddings.

During the self-attention, the image latent $z_t$ is split into two latents: 1) a base latent $z_t^{base}$, and 2) an object-background latent $z_t^{ob}$. Given object masks $\{m_i\}_{i=1}^{K}$ and background mask $m_{bg}$, we segment $z_t^{ob}$ into separate objects and background latents:

$$\{z_t^{o_i}\}_{i=1}^{K} = \text{Segment}(z_t^{ob}, \{m_i\}_{i=1}^{K}), \quad z_t^{bg} = \text{Segment}(z_t^{ob}, m_{bg}).$$

Each object latent $z_t^{o_i}$ and its corresponding prompt embedding $y_{o_i}$ are concatenated and passed through a Joint Self-Attention (JSA) module:

$$q_t^i = [(W_{qy} \cdot y_t^{o_i}); (W_{qz} \cdot z_t^{o_i})], \quad k_t^i = [(W_{ky} \cdot y_t^{o_i}); (W_{kz}.z_t^{o_i})], \quad v_t^i = [(W_{vy} \cdot y_t^{o_i}); (W_{vz} \cdot z_t^{o_i})],$$

$$[y^{o_i}, z^{o_i}] \leftarrow \text{Softmax}(\frac{(q_t^i)(k_t^i)}{\sqrt{d}}) \cdot v_t^i$$

where $W_{\cdot y}$ project prompt embeddings and $W_{\cdot z}$ project image latents. For simplicity, we reuse the same notation for the input and output of the transformer layer. This spatial-controlled self-attention is applied at each transformer layer, enabling precise control over object placement and appearance while preserving global visual consistency. The same mechanism is applied to the background: $[y_t^{bg}, z_t^{bg}] \leftarrow \text{JSA}(y_t^{bg}, z_t^{bg})$. The original base attention is applied on the base prompt and the base latent embeddings $[y_t^{base}, z_t^{base}] \leftarrow \text{JSA}(y_t^{base}, z_t^{base})$.

After the last transformer layer, the object and background latents are denoised from $t \to t-1$. The updated object latents $\{z_{t-1}^{o_i}\}_{i=1}^K$ and background latent $z_{t-1}^{bg}$ are then composed back into $z_{t-1}^{ob}$ using the segmentation masks.

Finally, we merge the base latent and object-background latent with a weighted sum: $z_{t-1} = z_{t-1}^{base} * \text{ratio}_{base} + z_{t-1}^{ob} * (1 - \text{ratio}_{base})$. It balances global coherence from the base latent and fine-grained spatial control from the object-background latent. We apply the spatial control for the initial $N_{sc}$ denoising steps.

## 4 EXPERIMENTS

### 4.1 EXPERIMENTAL SETUP

**Evaluation benchmarks.** We evaluate our method on T2I-CompBench (Huang et al., 2023) and NSR-1k (Feng et al., 2023b) datasets. They contain prompts rich in spatial, 3D, numeric, generally complex and surreal compositions. We evaluate our method on four categories from T2I-CompBench: 2D Spatial, Numeracy (Count), 3D Spatial, and Complex, each containing 300 prompts. For NSR-1K, we report results on the Spatial (283 prompts) and Count (672 prompts) categories.

**Evaluation metrics.** For the 2D-spatial and numeracy categories, we follow the standard evaluation protocols from T2I-CompBench and NSR-1k. Object detectors are used to identify, count, and measure spatial relations. For the 3D-spatial category, the original T2I-CompBench metric relies on outdated depth and detection models, resulting in unreliable scores. To address this, we introduce an MLLM-based metric aligned with recent evaluation standards (Zhang et al., 2023b) using GPT-4.1 (OpenAI, 2025). The model is prompted to identify all required objects and assess their 3D spatial relations. The final score is normalized to 0–100 and averaged over all examples. Further details on the limitations of the original metric and our new metric are provided in Appendix F. For the complex category, we adopt the 3-in-1 metric from T2I-CompBench, which averages the CLIP similarity score, spatial accuracy (via object detection), and BLIP-VQA accuracy. This composite score better aligns with human judgment.

**Implementation details.** We use GPT-4.1 (OpenAI, 2025) for LLM planning and SD3-Medium (SD3-M) (Esser et al., 2024) as the base diffusion model if not otherwise specified. We fix total 28 steps for denoising. The generation process is controlled by two key hyper-parameters: (1) $t_p$ – The time at which noise is sampled and applied to the prior image in the forward diffusion. As $t_p$ goes from (T to 0), prior strength increases, which increases faithfulness while reducing generative flexibility. (2) $N_{sc}$ – The number of steps for spatially controlled denoising. A higher value enforces stronger spatial control. The two hyper-parameters enable highly controllable generation and can be tuned to balance the composition performance and image quality, as demonstrated in Appendix A. For the experiments in this section, we sample $t_p$ corresponding to a high noise of 91.3% from the Flow matching schedule and set $N_{sc} = 3$ steps.

### 4.2 ABLATIONS

**Object prior quality.** The correctness of object priors is crucial for high quality image generation. We use the same metrics for evaluating the final image to evaluate the generated composite object prior. As shown in Table 1, the performance of the object priors is closely correlated with that of the final generated images. Notably, our prior-guided diffusion is applied only during the initial denoising steps. This design preserves the structural benefits of the prior while allowing subsequent steps to introduce generative flexibility,

Table 1: Performance of Object Priors (OP) and the corresponding Final Image (FI) on T2I-Compbench.

|    | Spatial | Numeracy | 3D | Complex |
|----|---------|----------|-------|---------|
| OP | 45.79   | 68.06    | 86.71 | 35.04   |
| FI | 48.24   | 68.21    | 77.16 | 38.66   |

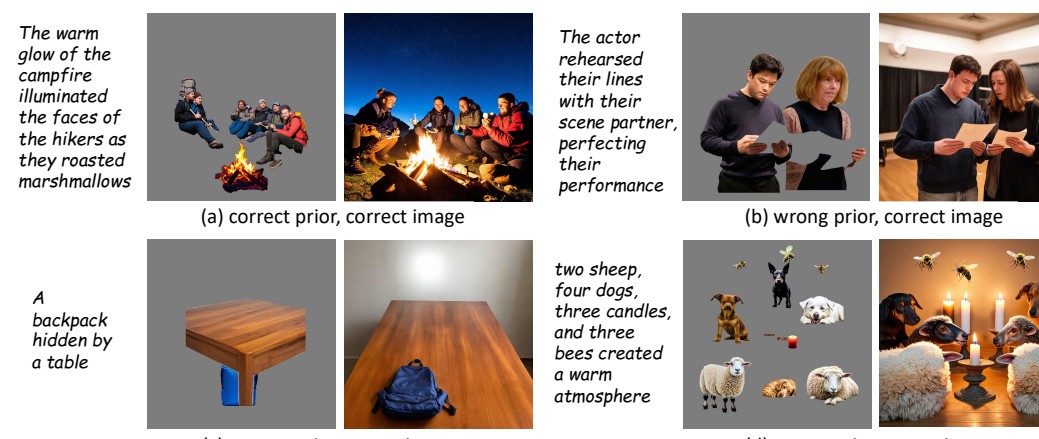

Figure 5: LLM generated object prior and their corresponding final image generation.

which helps correct inaccuracies in the initial composite. As illustrated in Figure 5(b), the final images can resolve issues present in the priors such as missing elements, incorrect orientation, size, and overall incoherence of the composite. Composition correctness and image quality/coherence can be easily balanced by tuning the hyper-parameters as shown in Appendix A. More detailed human evaluations on the quality of prior image and the corresponding generation is presented in Appendix B. Moreover, the correlation between the prior and the generation suggests that manually refining the prior can enhance performance, enabling interpretable generation with a human in the loop. We validate this on a few hard examples in Appendix C which show significant improvement.

**Prior-guided diffusion.** Table 2 shows the contribution of object prior reinforcement and spatial-controlled denoising in the prior-guided diffusion process. Each component significantly enhances performance over the base SD3-M model (Esser et al., 2024). Specifically, the object prior reinforcement yields absolute gains of over 14 and 6% in spatial and numeric categories, respectively, while spatial-controlled denoising improves performance by over absolute 12 and 4%. When combined, these components further boost results, demonstrating their complementary roles in achieving precise and controlled text-to-image generation.

Table 2: Impact of object prior reinforcement and spatial-controlled denoising on T2I-CompBench.

| Prior Reinforce | Spatial Control | 2D | Numeracy | 3D | Complex |
|---|---|---|---|---|---|
| ✗ | ✗ | 31.32 | 60.22 | 49.43 | 37.71 |
| ✓ | ✗ | 45.33 | 66.08 | **77.38** | 38.16 |
| ✗ | ✓ | 43.56 | 64.42 | 75.48 | 37.41 |
| ✓ | ✓ | **48.24** | **68.21** | 77.16 | **38.66** |

**Improvement over base models.** We integrate our method on architecturally diverse models with training paradigms, spanning both diffusion and flow-matching frameworks, including the U-Net–based SDXL (Podell et al., 2023), the Transformer-based SD3-M (Esser et al., 2024) and FLUX (Black Forest Labs, 2024). All methods are evaluated under the same object prior and layout to ensure comparability. As shown in Table 3, our approach yields significant improvements over all base models, underscoring its ability to generalize across fundamentally different architectures.

Table 3: Performance of different base models, with ComposeAnything consistently improving results.

| Method | T2I-CompBench | | | |
|---|---|---|---|---|
| | 2D-Spatial | Count | 3D-Spatial | Complex |
| SDXL | 21.33 | 49.88 | 47.12 | 32.37 |
| + *ComposeAnything* | **44.64** | **57.22** | **71.03** | **36.20** |
| SD3 | 31.32 | 60.22 | 49.43 | 37.71 |
| + *ComposeAnything* | **48.24** | **68.21** | **77.16** | **38.66** |
| FLUX | 26.13 | 60.58 | 59.51 | 37.03 |
| + *ComposeAnything* | **44.21** | **67.83** | **76.14** | **37.36** |

### 4.3 COMPARISON TO STATE OF THE ART

**Inference-based methods.** We compare our method against state of the art inference-based approaches, including general pretrained T2I models (SDv1 (Rombach et al., 2022), SDXL (Podell et al., 2023), SD3-M (Esser et al., 2024), and FLUX (Black Forest Labs, 2024)), layout-guided

Table 4: Comparison for inference-based methods on the T2I-CompBench and NSR-1k benchmarks.

| Method | T2I-CompBench | | | | NSR-1K | |
|---|---|---|---|---|---|---|
| | 2D-Spatial | Count | 3D-Spatial | Complex | Spatial | Count |
| SD-v1 (Rombach et al., 2022) | 12.46 | 44.61 | – | 30.80 | 16.89 | 31.45 |
| Attend-Excite v2 (Chefer et al., 2023) | 14.55 | 47.67 | – | 34.01 | 26.86 | 39.41 |
| SDXL (Podell et al., 2023) | 21.33 | 49.88 | 47.12 | 32.37 | 31.57 | 30.62 |
| RealCompo (Zhang et al., 2024b) | 31.73 | 65.92 | – | – | - | - |
| SD3-M (Esser et al., 2024) | 31.32 | 60.22 | 49.43 | 37.71 | 44.43 | 44.61 |
| FLUX (Black Forest Labs, 2024) | 26.13 | 60.58 | 59.51 | 37.03 | 39.29 | 55.97 |
| RPG (Yang et al., 2024) | 40.26 | 56.39 | 50.93 | 36.53 | 52.24 | 39.81 |
| Inference-scale (Ma et al., 2025) | 31.51 | 67.89 | – | 38.10 | - | - |
| **ComposeAnything (Ours)** | **48.24** | **68.21** | **77.16** | **38.66** | **63.80** | **59.36** |

training-free approaches (RPG (Yang et al., 2024) and RealCompo (Zhang et al., 2024b)), and a noise search method (inference time scaling (Ma et al., 2025)). As shown in Table 4, our method achieves the best performance on all the metrics, consistently adhering to the input prompt.

To further assess image coherence and perceptual quality, we conduct human evaluations. The rater annotation guidelines are detailed in Appendix D. We compare with the state-of-the-art method RPG (Yang et al., 2024). As shown in Figure 6, human raters consistently prefer our method over RPG across all metrics. Qualitative comparisons in Figure 7 further highlight that baselines such as SD3-M (Esser et al., 2024) and RPG (Yang et al., 2024) often fail to follow the input prompt. In particular, they struggle with challenging spatial relations (e.g., a chicken behind the clock) and maintaining the correct object counts (e.g., number of giraffes, microwaves).

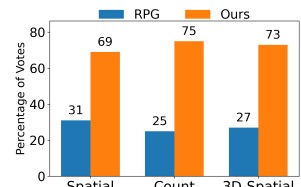

Figure 6: Human evaluations against inference-based method RPG.

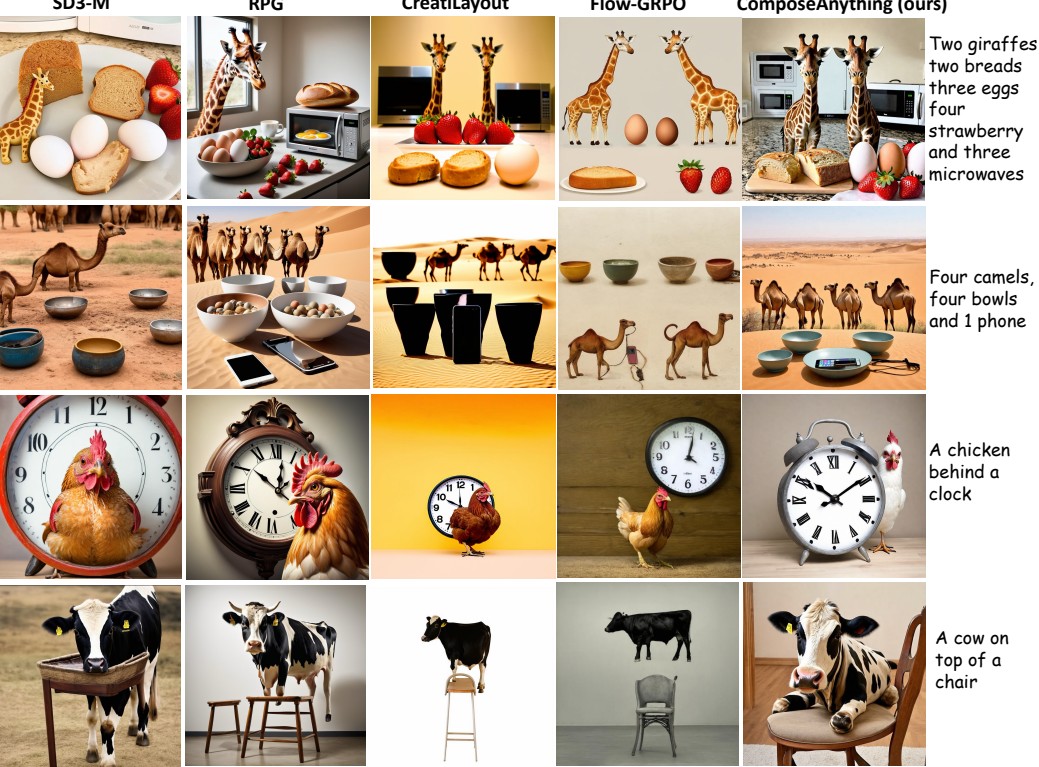

Figure 7: State-of-the-art comparison against SD3-M (Esser et al., 2024), RPG (Yang et al., 2024), Creatilayout (Zhang et al., 2024a), and Flow-GRPO (Liu et al., 2025) on complex surreal prompts.

Table 5: Comparison against training methods on the T2I-CompBench and NSR-1k

| Method | T2I-CompBench | | | | NSR-1K | |
| --- | --- | --- | --- | --- | --- | --- |
| | 2D-Spatial | Count | 3D-Spatial | Complex | Spatial | Count |
| LayoutGPT (Feng et al., 2023b) | 45.81 | 60.27 | – | – | 60.6 | 55.6 |
| CreatiLayout (Zhang et al., 2024a) | 47.36 | 62.15 | 68.85 | 34.6 | 59.8 | 63.4 |
| Flow-GRPO (Liu et al., 2025) | **54.47** | 67.52 | 48.49 | 38.42 | **74.09** | **65.27** |
| **Training-free** | | | | | | |
| ComposeAnything (Ours) | 48.24 | **68.21** | **77.16** | **38.66** | 63.80 | 59.36 |

**Training-based methods.** The compared training-based methods include layout-to-image models with box conditioning such as Gligen (Li et al., 2023a), CreatiLayout (Zhang et al., 2024a), and Reinforcement learning method like Flow-GRPO (Liu et al., 2025).

Quantitative comparisons with automatic metrics are presented in Table 5. Although training-based methods outperform our training-free method in some categories where they were optimized for these metrics, they tend to change the real image distribution to fit the surreal image prompts, which significantly compromise the image quality, resulting in floating objects,

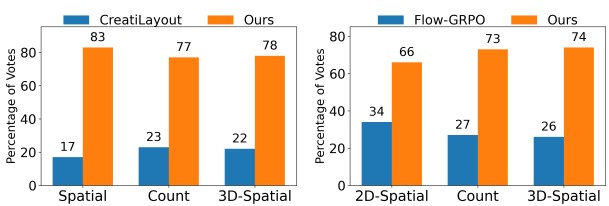

Figure 8: Human evaluations against training-based methods CreatiLayout and Flow-GRPO.

faded background and broken physics, as shown in Figure 7. Therefore, we further conduct human evaluations (Figure 8), where our method is consistently preferred over both Flow-GRPO and CreatiLayout. By leveraging object priors and integrating them into the denoising process, our method achieves a better balance between compositional fidelity and visual quality.

**Efficiency.** We provide computation time comparisons in Table 6. We compute the time over prompts with 3, 5, 7, 10 objects and average over all. For fair comparison, we use the same LLM planning for CreatiLayout and our method. As RPG uses region crops instead of object layout, we use their original LLM planning module, which is less efficient than ours. For prior generation, we generate and segment all

Table 6: Runtime breakdown (seconds).

| Method | LLM planning | Prior generation | Image generation | Total time |
| --- | --- | --- | --- | --- |
| CreatiLayout (SD3-M) | 4.98 | – | 4.92 | 9.90 |
| Ours (SD3-M) | 4.98 | 4.50 | 5.62 | 15.1 |
| RPG (SDXL) | 9.14 | – | 12.98 | 22.11 |
| Ours (SDXL) | 4.98 | 4.50 | 4.97 | 14.45 |

objects in parallel using multiple GPUs to reduce the computation time, allowing scalability. For image generation, our method applies prior reinforcement and SCD only for the first few steps (6 and 3 respectively out of 28 steps). This leads to faster overall generation compared to RPG, which applies regional diffusion throughout all steps.

## 5 CONCLUSION

In this work, we introduce *ComposeAnything*, a novel inference-time framework for compositional text-to-image generation that leverages object-level guidance derived from LLM-generated 2.5D semantic layouts. By introducing a composite object prior for structured initialization and prior-guided diffusion, our approach enables precise object placement and robust semantic grounding without any additional training. *ComposeAnything* achieves state-of-the-art performance on T2I-CompBench and NSR-1K, effectively balancing image quality and prompt fidelity even under complex or surreal scenarios. Our results highlight the potential of LLM-driven reasoning and composite prior guidance in advancing compositional T2I generation.

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

APPENDIX

We start with the analysis of the impact of two key hyper-parameters in our framework in Section A. Section B provides more detailed results of the correctness of object prior and final images, including human evaluation and qualitative examples. Section C demonstrates potential for human-in-the-loop manual prior correction for better generation. Section D introduces the human evaluation details for state-of-the-art comparison. Section E presents the LLM prompt for 2.5D semantic layout generation and illustrates an output example. Section F details the LLM prompt for evaluating the 3D-Spatial category. Finally, Section G describes the use of LLMs in our work.

## A  IMPACT OF KEY HYPER-PARAMETERS

We analyze the effect of two key hyper-parameters $t_p$ and $N_{sc}$, as discussed in Section 4.1 of the main paper. $t_p$ is the time at which noise is sampled and applied to the prior image in the forward diffusion. It controls the object prior reinforcement strength (OPR). Lower values denote stronger priors. $N_{sc}$ is the number of steps for spatially controlled denoising. It controls the spatial-controlled denoising strength (SCD). Larger values result in stronger control.

Figure 9 presents object priors and corresponding generated images for two text prompts, under varying $t_p$ and $N_{sc}$ values. In the first row of each example, $N_{sc}$ is fixed at 3 to isolate the impact of OPR with different $t_p$. At low OPR strength, the final image fails to preserve the *appearance and semantics* of the prior, for example, the butterfly appears in front of the cup not on top of it, and the number of clocks and microwaves is incorrect. As the strength of OPR increases, objects' semantics and appearance such as color, shape and number are more strongly retained. However, excessive reinforcement reduces generative flexibility, leading to over-constrained and less natural outputs.

In the second row of each example, $t_p$ is fixed at 0.91 to examine the effect of $N_{sc}$. Low SCD strength leads to limited spatial control, with objects leaking in background (extra cup), objects getting merged (both dogs merged), and incorrect object counts. As we increase the SCD strength, object positions and sizes from the prior are more faithfully preserved in the final image. However, too strong spatial control results in low-quality compositions such as rigid placements, incoherent scene, floating objects similar to training-based box-conditioned methods (Zhang et al., 2024a).

Therefore, in our experiments, we set $t_p = 0.91$ and $N_{sc} = 3$ to strike a balance between faithful prompt adherence, generative flexibility, and overall scene coherence. Both hyper-parameters are beneficial and complementary to reliably produce correct spatial relations, accurate object counts and high-quality images.

## B  DETAILED EVALUATION OF OBJECT PRIORS AND FINAL IMAGES

To better assess the quality of object priors and their influence on final image generation, we categorize results into four combinations: i) correct prior – correct image, ii) correct prior – incorrect image, iii) incorrect prior – correct image, iv) incorrect prior – incorrect image.

We conduct a human evaluation using 30 samples per category, i.e., pairs of the prior and final image, across the 2D-Spatial, 3D-Spatial, and Numeracy categories in T2I-Compbench. For each sample, annotators perform a 4-way classification task, judging the correctness of both the prior and the resulting image. The annotation interface is illustrated in Figure 11.

Figure 10 presents the results of the human evaluation. As seen in the bar plots for the 2D and 3D-Spatial categories, the majority of samples fall into the correct-prior and correct-image category. In contrast, the Numeracy category shows a higher occurrence of correct-prior but incorrect-image cases. This is primarily due to the increasing number of objects, where relative object sizes in the prior affect its realism and quality. While our model's generative flexibility helps correct such visual artifacts during image synthesis, it often does so at the expense of count accuracy, leading to mismatches in object quantities, as illustrated in Figure 11 (bottom). Moreover, as the number of objects increases, the priors tend to become less coherent overall, resulting in a greater frequency of incorrect-prior and incorrect-image cases.

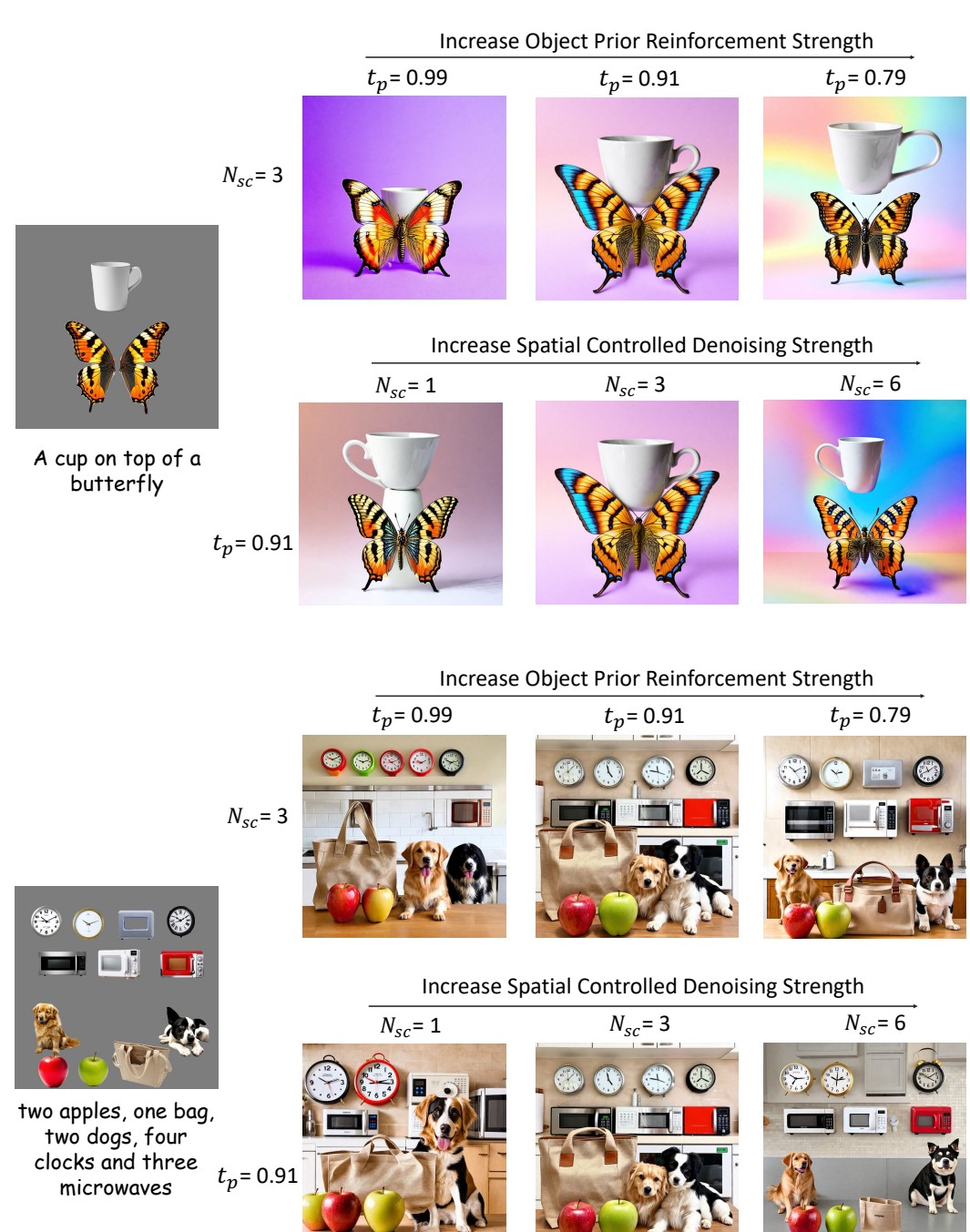

Figure 9: Effect of Object Prior Reinforcement and Spatial-Controlled Denoising. Increasing either strength enhances appearance fidelity and spatial precision, but reduces generative flexibility.

Figure 12 - 15 provide more examples of the object priors and corresponding final images on T2I-Compbench dataset, including 2D-spatial, 3D-spatial, non-spatial, numeracy and complex prompt categories.

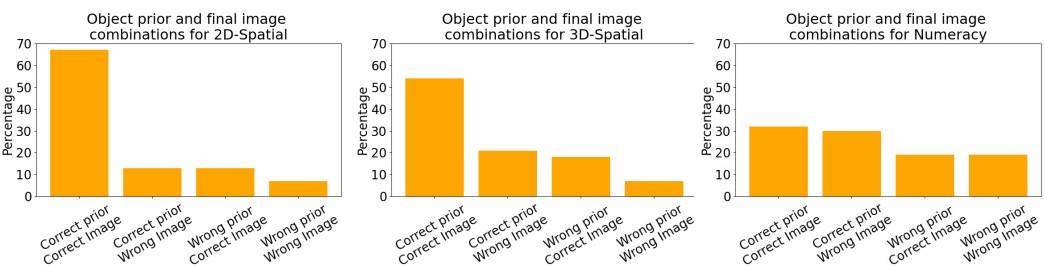

Figure 10: Human evaluation results on the correctness of prior and final image pairs on the three categories of T2I-Compbench dataset.

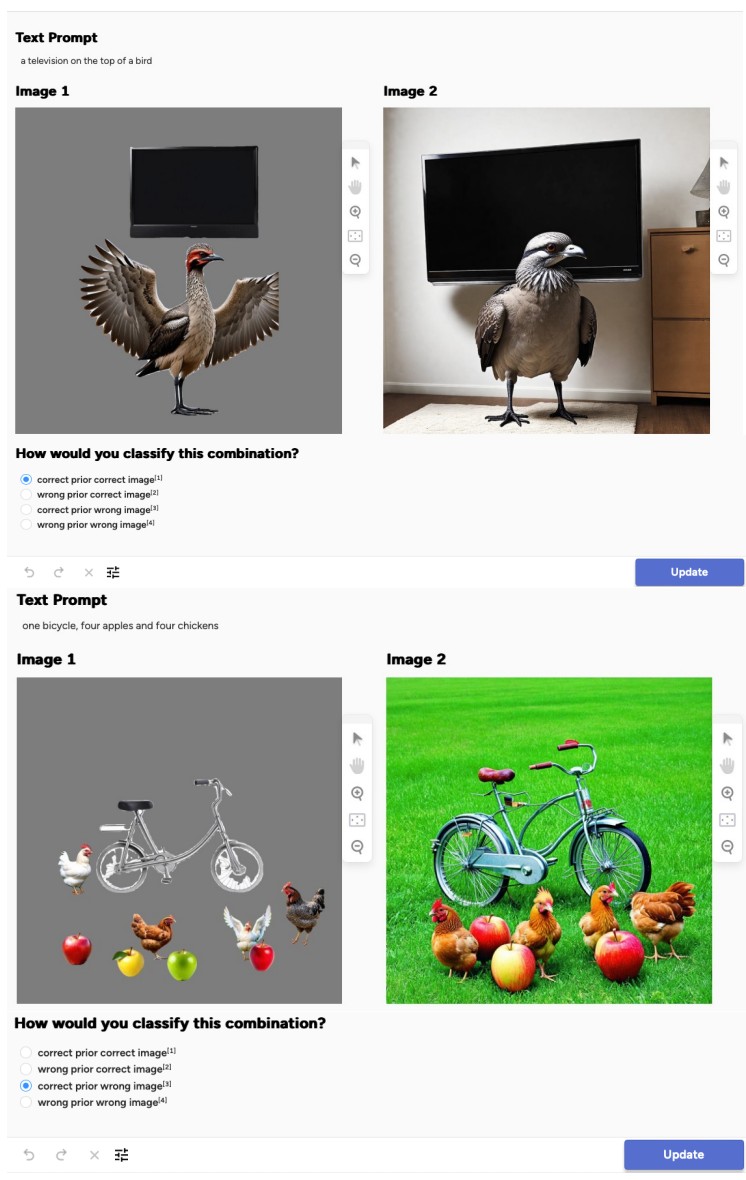

Figure 11: Labeling interface for evaluating object prior and the final image.

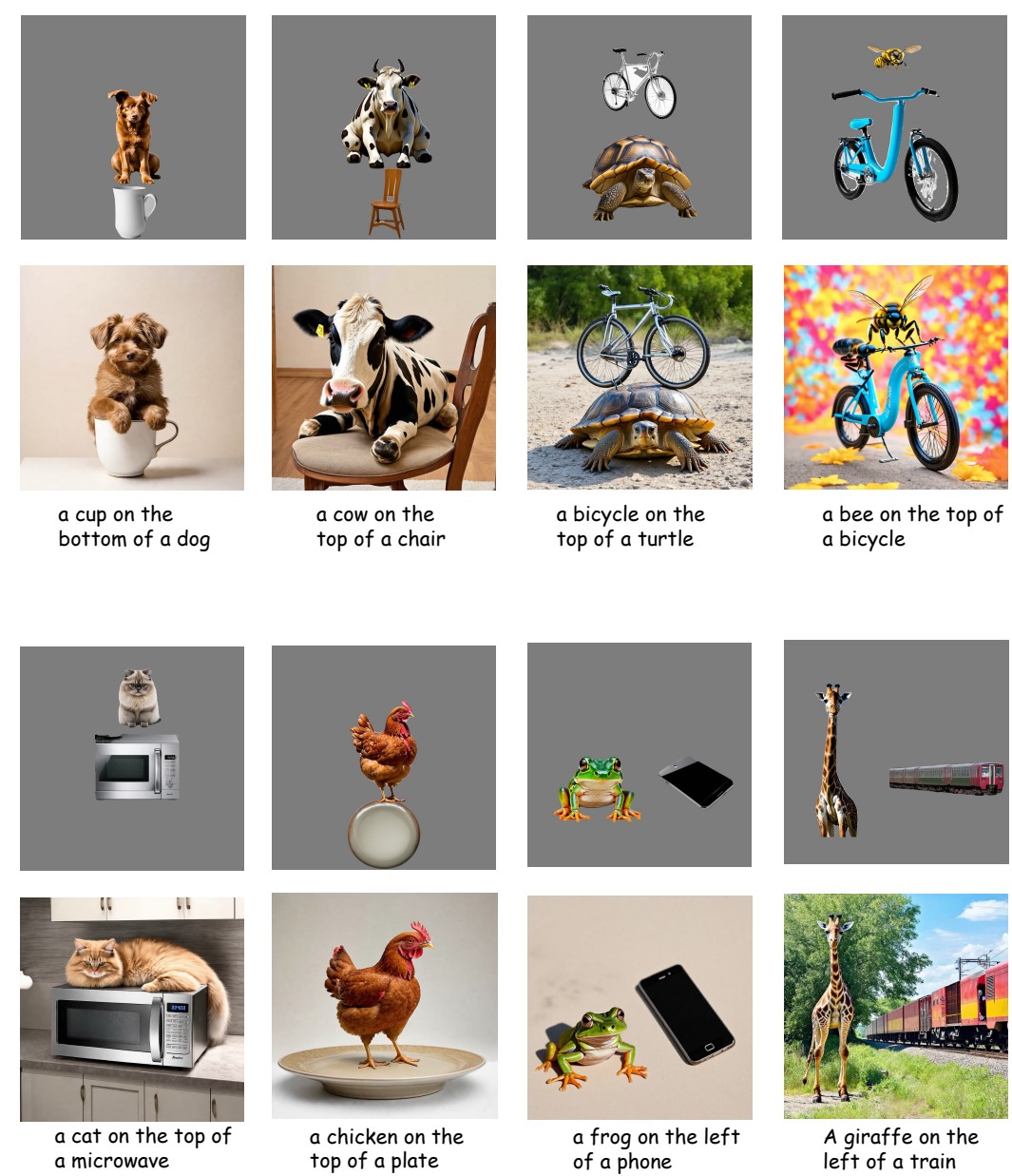

Figure 12: Object prior and the corresponding generation for 2D-Spatial compositions from T2I-compbench.

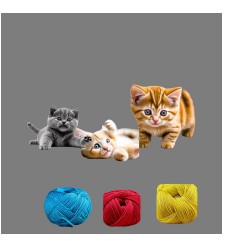
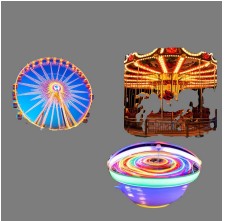
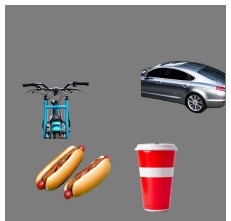
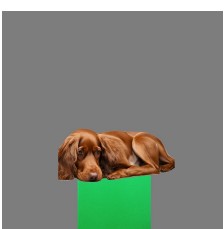

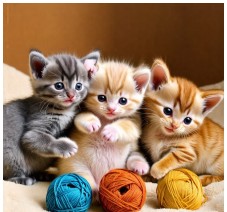
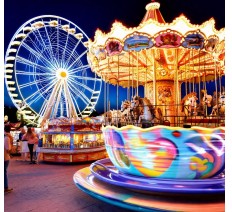
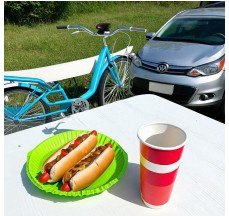
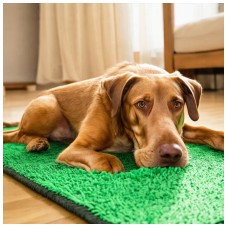

The soft, furry kittens played together in a pile on the warm, cozy blanket, their tiny paws batting at colorful balls of yarn

The vibrant, glittering lights of the carnival rides spun and twirled in dizzying circles, thrilling and delighting the adventurous

Two hot dogs sit on a green paper plate near a soda cup which are sitting on a white picnic table while a bike and a silver car are parked nearby.

The brown dog was lying on the green mat

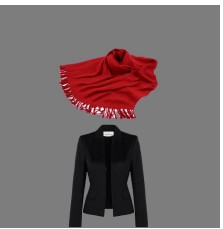
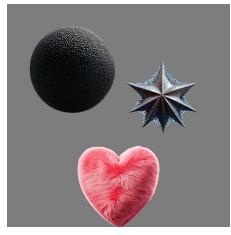
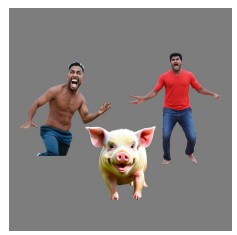
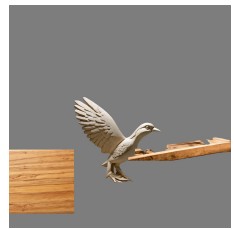

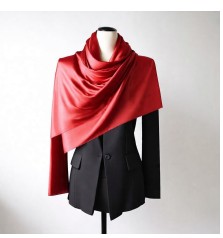
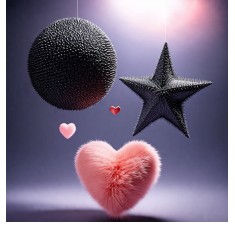
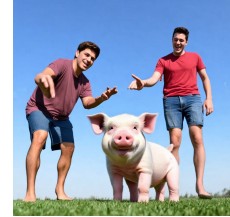
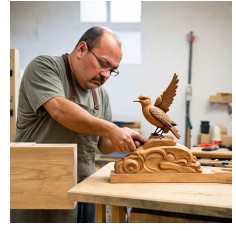

The long red scarf draped over the short black jacket.

The bumpy sphere was suspended in mid-air next to the spiky star and the fuzzy heart.

two men and one pig played in the yard

The woodcarver is creating a sculpture of a bird from a block of wood

Figure 13: Object prior and the corresponding generation for Complex compositions from T2I-Compbench.

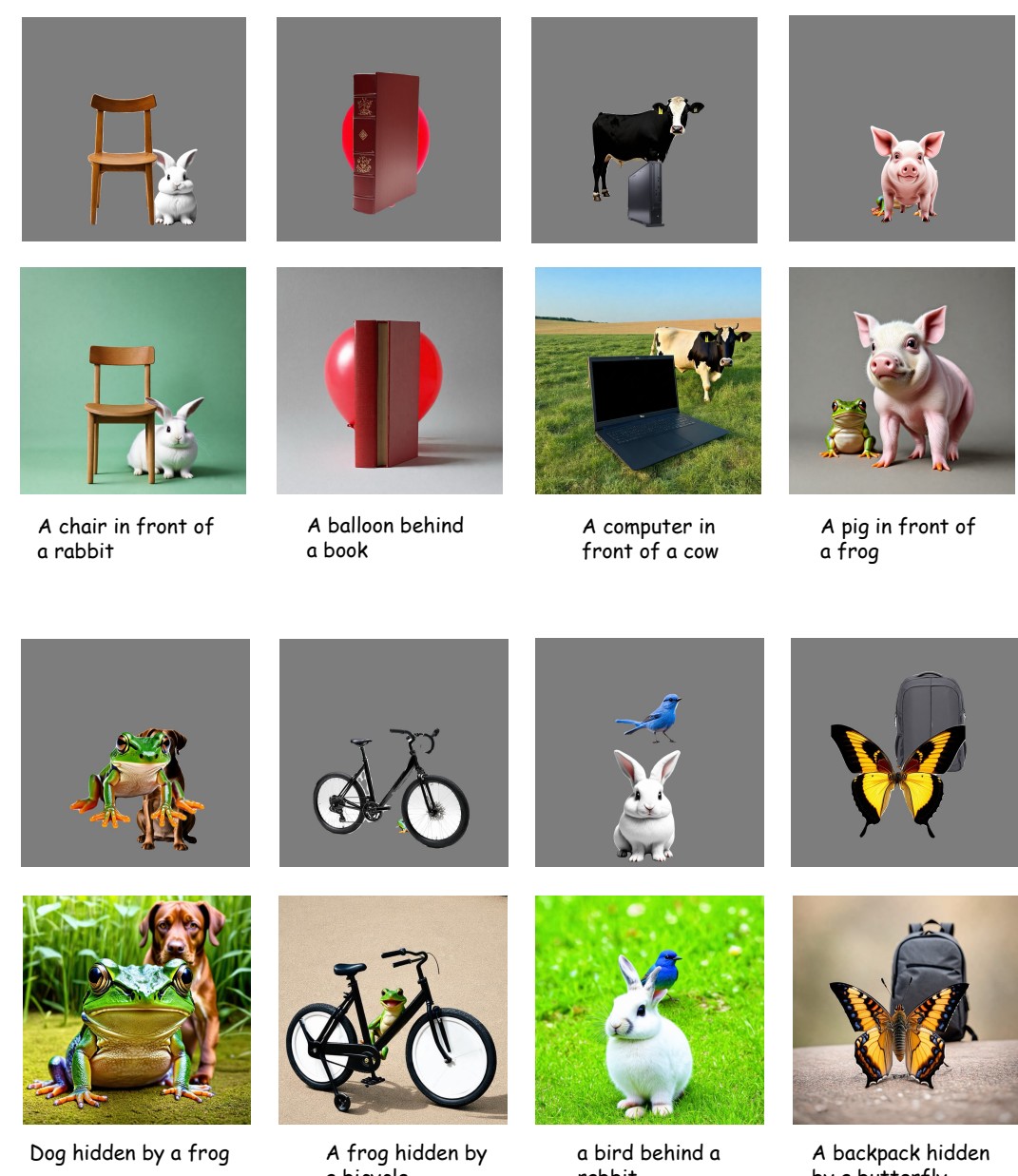

A chair in front of a rabbit

A balloon behind a book

A computer in front of a cow

A pig in front of a frog

Dog hidden by a frog

A frog hidden by a bicycle

a bird behind a rabbit

A backpack hidden by a butterfly

Figure 14: Object prior and the corresponding generation for 3D-Spatial compositions from T2I-compbench.

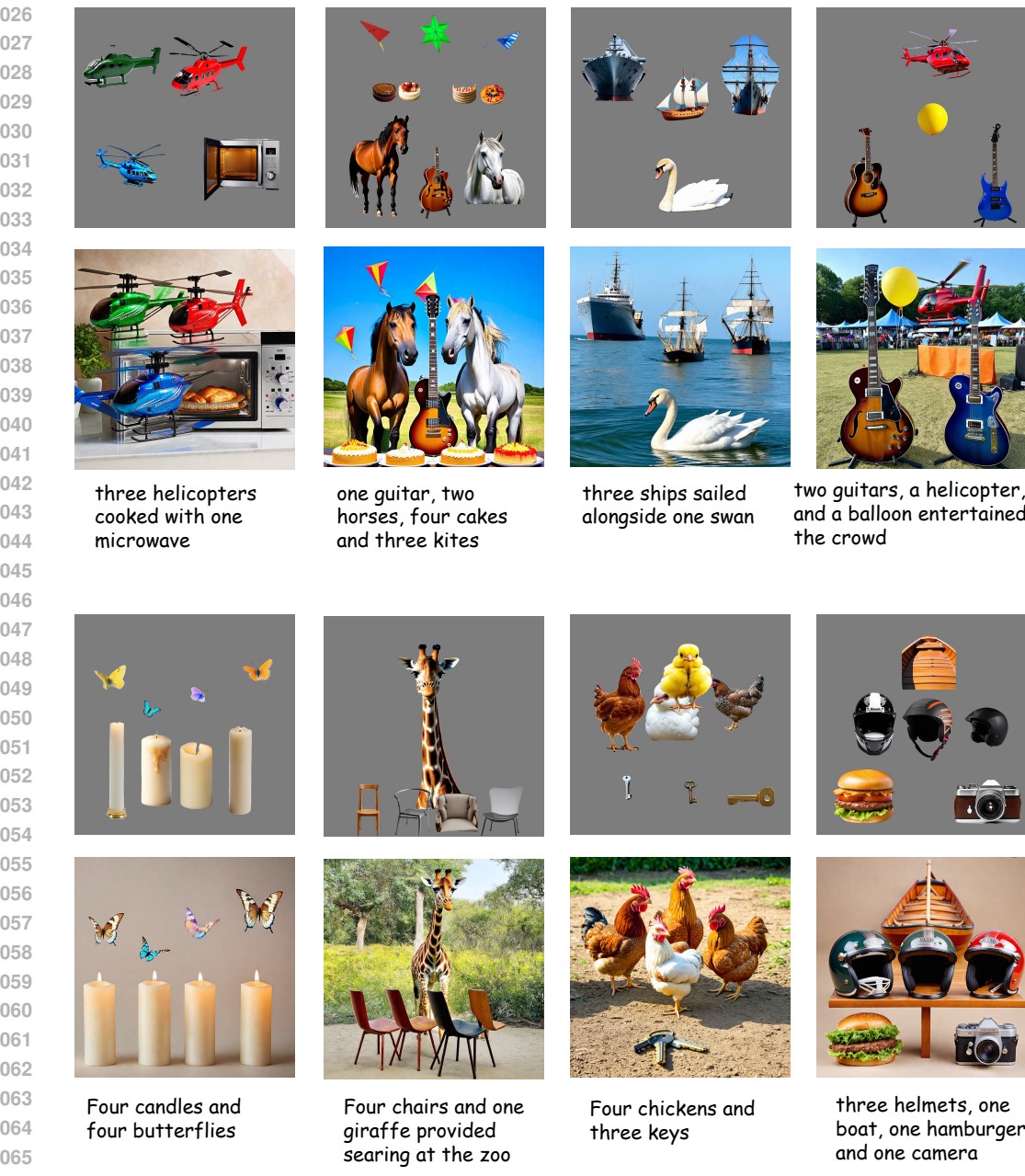

Figure 15: Object prior and the corresponding generation for Numeracy compositions from T2I-compbench.

## C    MANUAL PRIOR ADJUSTMENT

Our framework enables interpretable generation with multiple options for human in the loop to improve object priors, such as editing intermediate-generated LLM layouts, modifying object and background captions, and selecting the most suitable objects for the composed prior image. Better prior images significantly enhance the quality of final images, making human guidance a powerful tool for improving overall results. Figure 16 shows by simply changing the positions and resizing the objects in the prior image, our method can correct generated image.

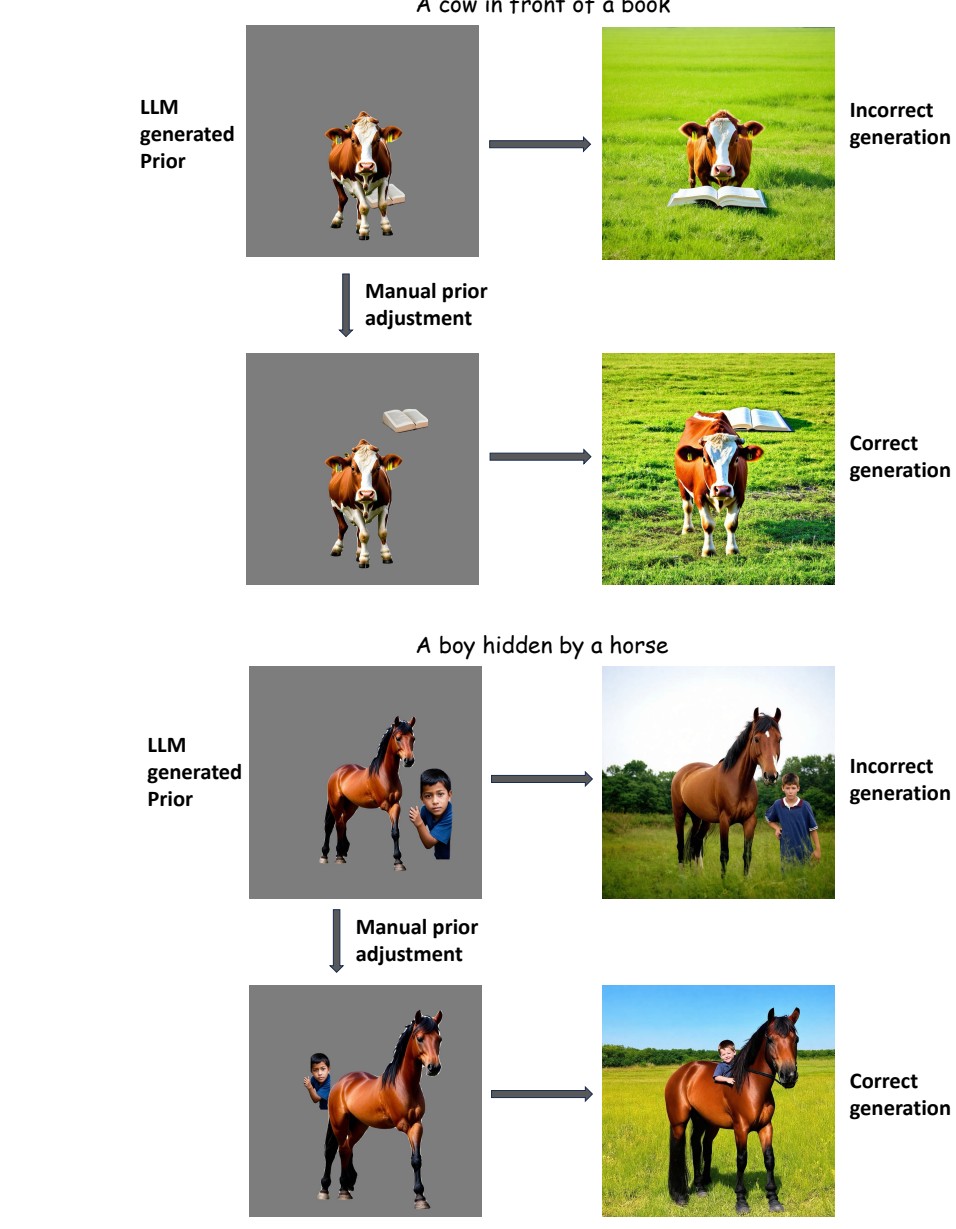

Figure 16: Manual prior adjustment

# D    LABELING INTERFACE FOR HUMAN EVALUATIONS

We conduct human evaluations against the SOTA methods on three categories in T2I-CompBench: 2D-Spatial, 3D-Spatial and Count. For each category, we randomly sample 30 prompts and perform pairwise comparisons. Five raters participated in the evaluation, with 30% of images overlapping across raters to measure inter-annotator agreement. The average agreement scores were around 80% for all categories.

Figure 17 shows the human evaluation instructions and interface. The instructions focuses on prioritizing both correctness to prompt focusing on 2D-3D spatial relations and object count. Also to select images with higher quality. As can be seen in the "six horses" image, both images have 6 horses, but the first image has better quality.

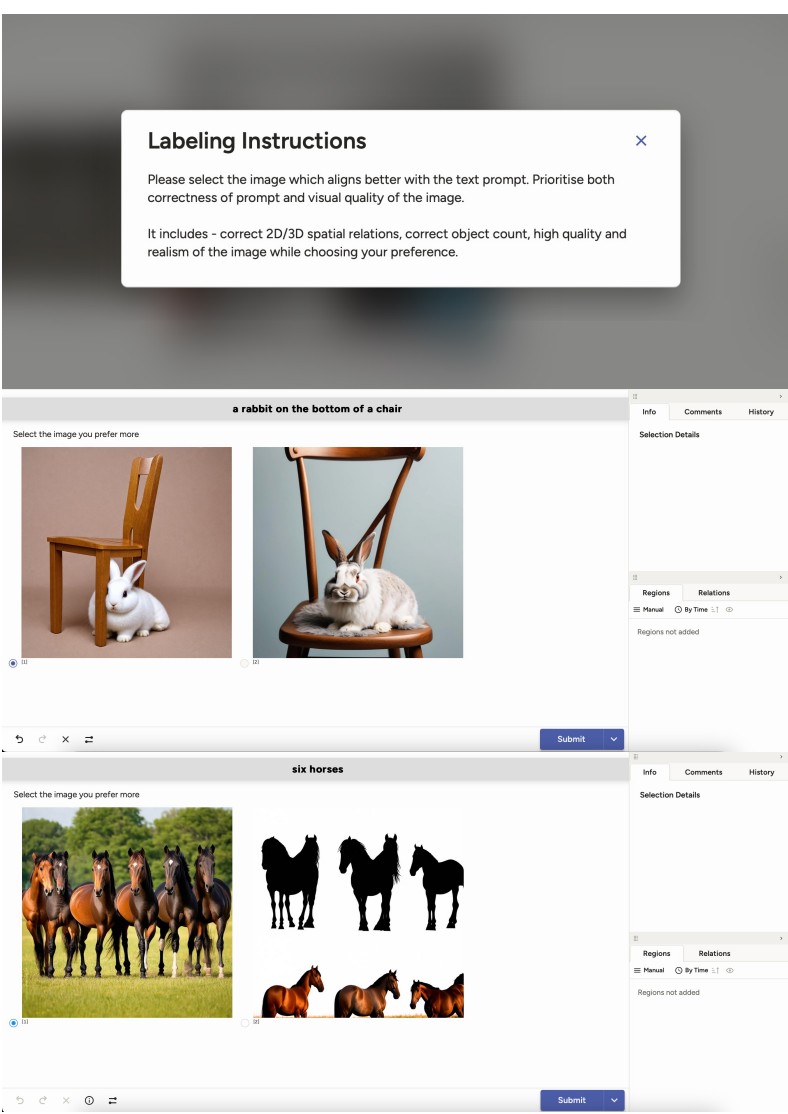

Figure 17: Labeling interface for human evaluations.

# E  LLM PLANNING INSTRUCTIONS

Figures 18 - 22 show the detailed instructions for LLM planning.

Figure 23 shows the output from the LLM for an example alongside the prior and final image.

---

You are a master of composition who excels at extracting key objects, their counts, attributes, and 2D and 3D spatial relationships from input text. You supplement the original text with imaginative details to create visually compelling layouts that adhere to beautiful aesthetics.

**Objective**:
Given a concise image prompt, plan the layout, extract all key objects, their positions, relative depth and generate captions to guide compositional image generation. The task involves 8 steps:
1. Plan the layout focusing on foreground object selection, plausibility to put the objects in a layout for a coherent image generation, location, orientation, size, depth, background etc.
2. Correct the language of the input prompt and rewrite it in simple and easy words.
3. Extract only foreground solid objects as planned and list down each object one by one
4. Create individual object caption that provide isolated visual details with no information leak from other objects. Strictly no mention of other objects in any way.
5. Produce relative depth for each object.
6. Produce bounding boxes for each object.
7. Generate a compositional image caption that describes the entire scene and respects the spatial-relation, count and, attributes, of the objects.
8. Generate an isolated one word/phrase background prompt.

**Rules and Guidelines:**
**1. Plan the layout:**
    a. Identify foreground objects that can be easily placed within a 2D box layout without severe entanglements.
       - Examples: "3 rabbits and 4 deers near a car" → all the 3 rabbits, 4 deers and, the car can be separately placed at 8 distinct locations in the layout.   "A pig on top of a man" → A pig can be easily placed on top of a man as two individual boxes in a 2D layout. "A table placed on a rug" → The two objects overlap, but can be easily placed in a 2D layout, as there are no severe 3D entanglements.
    b. Entangled Object Interactions: When objects are part of each other, or are too entangled they should be treated as a single object.
       - Examples: A woman wearing a ring → Extract only woman, instead of separate woman and ring objects. A man throwing a basketball → Extract only man, because it is hard to accurately place the ball in the hands on the players in simple 2D layouts, so consider the two objects as a single entity. "A woman in white shirt and black jeans" → Extract only Woman as it is hard to disentangle clothes from the person.
    c. Keep the original counts, spatial positions, attributes intact.
    d. If exact spatial relations, count, color etc are not provided deduce from it. For example: "four bears and four sofas and less number of cats". Deduce the number of cats, in this case it could be one or two. And the cats could be in foreground near to the viewer, lying down together. Bears in the midground sitting or playing, and sofas arranged neatly in the back.
    e. Do not extract elements that describe the environment, background, or intangible phenomena that can't be put in box layout. Example exclude background features like 'wall', 'floor', 'ceiling', 'bathroom', 'tiles', 'kitchen', 'room', 'field', 'grass', 'sky', 'river', 'forest', 'rain', 'sunset', 'snow', 'fog', 'wind', 'city', 'scent', 'fragrance', 'heat', 'fire', 'waterdrops', 'water', etc"
    f. Consider the 3D positioning, objects closer to viewer should appear a little bigger and the object far in the back should appear a little smaller.

**2. Rewrite the caption:**
    a. Correct the language if there are any mistakes.
    b. Do not reinterpret, reverse, or replace any meaning
    c. rewrite it based on the planning

---

Figure 18: Instructions for LLM planning (to be continued).

**3. Foreground object Extraction:**
   a. Extract only foreground objects without severe entanglements as planned.
   b. Strictly keep the original counts, and enumerate every object one at a time.
   c. Make sure to extract the accurate counts and enumerate every object one at a time.

**4. Object Descriptions (isolated objects):**
   a. Describe each object individually with strictly no information leak from other objects. Only focus on object's own attributes, and the way it interacts with the scene, but not other objects.
   b. Strictly Don't mention any detail of other objects or background, while describing the current object.
   c. Strictly no mention of other objects count, positions, attributes etc. in any way.
   d. Description should be very consise in one line.

**5. Relative Depth:**
   a. For every extracted object predict the relative depth of the object based on the given prompt or imagination. If there are a total of 3 objects, 2 in the foregroud and 1 behind them depth should be: object1: 1, object2: 1, object3: 2.

**6. Bounding Box Layout (Relative Sizes):**
   a. Scale bounding boxes to reflect any surreal or exaggerated proportions.
   b. Bounding boxes should be generally big, covering major parts of the image.
   c. Bounding boxes must be placed such that every object remains visibly distinct on a 2D canvas. Even when objects differ in depth, they must not fully occlude or obscure one another.
   d. For objects close to the viewer in the foreground size should be bigger than the objects in far back (to emphasize depth). Also consider the original size of the objects.

**7. Composition Caption (Scene-Level):**
   a. Craft a unified caption that highlights how all extracted objects interact within the scene. Objects should cover most of the part of the image. Mention a one phrase background.
   b. If the input caption is a surreal composition, then imagine the object interactions accordingly and compose a coherent scene.
   c. Emphasize object attributes, spatial relationships, and background.

**8. Background (isolated background):**
   a. Strictly mention a one phrase background, with no object information.
   b. If provided, extract the background information from the input caption. Example: A white door in a pink bathroom. Here pink bathroom should be the backround.

**9. Output Format rules:**
Use the following format strictly:
Object class heading: "Objects:"
      1. [object_name]
      2. [object_name]
   Object descriptions heading: "Object_Descriptions"
    For each object, use the following structure:
    Caption Heading: "Caption_object_n(isolated):"
    Depth Heading: "Relative_depth_n:"
 Bounding Box Heading: "Box_object_n:"
   Compositional caption heading: "Compositional_caption:"
   Background caption heading: "Background_caption:"
  c. Bounding boxes should include absolute positions in the format [x_top, y_top, x_bottom, y_bottom] for a 1024x1024 resolution canvas.
   d. Strictly both compositional caption and object captions should not be more than 70 words.

Figure 19: Instructions for LLM planning (to be continued).

**Examples:**

**Example 1:**
a green balloon on the bottom of a cat

**Planning**: "a green balloon on the bottom of a cat" could mean that the cat is sitting on a balloon in a surreal or a playful scene. Both the objects can be easily put in a 2D layout as they are not heavily entangled. Cat is sitting on a ballon so both have the same depth.

**Rewritten caption:**
a white cat is sitting on top of a green balloon

**Objects:**
1. green balloon
2. cat

**Object_Descriptions:**
Caption_object_1(isolated): A green balloon is lying on the floor.
Relative_depth_1: 1
Box_object_1: [300, 700, 724, 1024]
Caption_object_2(isolated): A white cat, sits calmly with its legs and tail folded.
Relative_depth_2: 1
Box_object_2: [200, 300, 824, 750]

**Compositional_caption:** A fluffy white cat, full of vibrant energy is sitting playfully on top of a blue balloon in a cosy room.
Background_caption: A cosy room

**Example 2:**
The soft, fluffy texture of the cotton candy melted in the mouth, a sugary treat of childhood nostalgia.

**Planning:** Since both the objects "boy" and "cotton candy" are interacting closely in an intrecate way "melted in the mouth", it is hard to put the two objects in a 2D layout as the object is in the mouth of the boy, there are occlusions and orientation complexities, qualifying for severe entanglement. So considering both the objects as one. The boy could be holding a cotton candy on his left and eating it.

**Rewritten caption:**
A boy is eating a soft fluffy cotton candy.
Objects:
1. boy

**Object_Descriptions:**
Caption_object_1(isolated): A little boy facing left is eating a pink cotton candy.
Relative_depth_1: 1
Box_object_1: [300, 150, 700, 874]

**Compositional_caption:** A little boy facing left eagerly enjoys a fluffy pink cotton candy in a lively park, his eyes sparkle with excitement.

**Background_caption**: a lively park

Figure 20: Instructions for LLM planning (to be continued).

**Example 3:**
two helmets and three ships

**Planning:** There are three ships and two helmets, each treated as individual, rigid objects to be placed independently in the scene. Given their differing scales, the ships (being much larger) should be positioned in the background, while the smaller helmets should be placed in the foreground.
Depth Assignment: Helmets: depth = 1 (foreground). Ships: depth = 2 (background)
Size Ratio: To reflect realistic scale in a 2D layout, the bounding boxes for helmets and ships should follow an approximate size ratio of 1:2, with ships appearing larger.
Placement Details: The helmets should be located near the bottom of the scene, close to the shore in the foreground. The ships should appear near the horizon, conveying depth and distance. Among the three ships: two can be large and face forward (toward the viewer), while the third, a smaller wooden ship, can be placed in the center, angled left to reveal more of its body and shape.

**Rewritten caption:** Three ships are sailing in the sea near the horizon, while two helmets are placed on the shore in the foreground

**Objects:**
1. ship
2. ship
3. ship
4. helmet
5. helmet

**Object_Descriptions:**
Caption_object_1(isolated): A metalic ship facing front.
Relative_depth_1: 2
Box_object_1: [50, 0, 350, 510]
Caption_object_2(isolated): A metalic ship facing front.
Relative_depth_2: 2
Box_object_2: [650, 90, 1000, 520]
Caption_object_3(isolated): A small wooden ship facing left.
Relative_depth_3: 2
Box_object_3: [234, 550, 824, 750]
Caption_object_4(isolated): A black helmet.
Relative_depth_4: 1
Box_object_4: [100, 790, 400, 1020]
Caption_object_5(isolated): A red helmet.
Relative_depth_5: 1
Box_object_5: [600, 780, 900, 1020]

**Compositional_caption:** Two majestic ships sail across the horizon with towering masts and curved hulls in the sea. One small wooden ship is in the middle. In the foreground, two helmets, one sleek black and the other bold red rest side by side on the shore.

**Background_caption:** a sea shore

Figure 21: Instructions for LLM planning (to be continued).

**Example 4:**
A plane is hidden by a blue frog

Planning: The frog and the plane are separate, non-entangled objects and can be laid out independently in a 2D composition. Since the frog hides the plane, the frog should be large and positioned in the foreground (depth 1), while the plane appears smaller in the background (depth 2). To preserve visibility, the frog is placed slightly left of center, and the plane is positioned more toward the right. The frog partially overlaps the left edge of the plane, but at least half of the plane remains clearly visible.

**Rewritten Caption:** a blue frog in the front hides a plane in the back

**Objects:**
1. blue frog
2. plane

Caption_object_1(isolated): A blue frog facing front sitting calmly

Relative_depth_1: 1

Box_object_1: [200, 300, 700, 900]

Caption_object_2(isolated): A commercial airplane on ground facing left

Relative_depth_2: 2

Box_object_2: [520, 350, 920, 550]

**Compositional_caption:** A frog is sitting in close in the foreground hiding a commercial airplane in far back

**Background_caption:** A runway

Figure 22: Instructions for LLM planning.

Input Prompt: The soft, warm glow of the campfire illuminated the faces of the hikers, as they roasted marshmallows and swapped stories.

**Planning:** The "faces of the hikers" are illuminated by the campfire, while they roast marshmallows and swap stories. Since roasting marshmallows requires close hand-object interaction, this counts as a heavy entanglement. The group of hikers is treated as a single foreground object, with the campfire as another. Both can be easily placed in a visually coherent 2D layout:, the campfire in the foreground (at depth = 1) the group of seated or standing hikers((at depth = 2) arranged in a semicircle behind it, both occupying significant portions of the image. Background is implied as outdoors at night, likely in a forest or clearing.

**Rewritten caption:**
A group of hikers sit around a glowing campfire, their faces lit warmly as they roast marshmallows and share stories.

**Objects:**
1. campfire
2. group of hikers

**Object_Descriptions:**
**Caption_object_1(isolated):** A glowing campfire with bright orange flames and flickering light.
**Relative_depth_1:** 1
**Box_object_1:** [350, 700, 674, 980]

**Caption_object_2(isolated)**: A group of hikers sitting and relaxing, with sticks, faces softly lit.
**Relative_depth_2:** 2
**Box_object_2:** [150, 350, 874, 850]

**Compositional_caption:** A warm campfire glows brightly in the evening, and a group of hikers sits closely around it, their faces softly illuminated. The hikers hold marshmallow sticks and enjoy stories together beneath the night sky.

**Background_caption:** a starry night outdoors

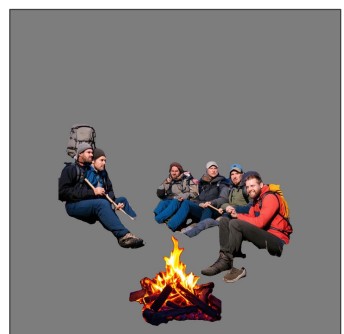 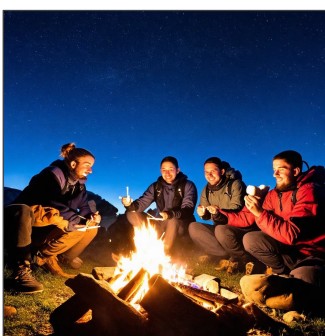

Figure 23: LLM planning for object prior generation, with final generated image.

# F  3D METRIC EVALUATION WITH LLM

For the 3D-spatial category, the original T2I-CompBench metric relies on depth estimation and bounding box detection, which we found inaccurate and overly punitive to the generated image. Evidence of these limitations appears in Table X (10) of the T2I-CompBench++ paper (Huang et al., 2023), where methods showing clear improvements in human evaluation achieve marginal or negative scores with the original metric. For example Attn-Exct+SDv2 vs. SDv2 shows +3.6 improvement in human evaluation but -0.08 in the original metric. To address this, we introduce an MLLM-based metric using GPT-4.1 (OpenAI, 2025). The model is prompted to first identify all required objects and then assess their 3D spatial relations. Scores are assigned as follows: 0 if objects are missing or 3D relations are wrong, 1 if all objects are present but the 3D relations are ambiguous, and 2 if everything is correct. We normalize the total score to a 0–100 scale and average over all examples.

Figure 24 presents the detailed instructions given to the LLM for evaluating 3D-spatial relations.

---

**Title**: Evaluate Spatial Relationships in the generated image.

**Objective**: Your task is to evaluate whether the spatial relationship described in the prompt is correctly represented in 3D space within the image.

**Prompt Example:**
"A red cube is in front of a green sphere."

**How to Evaluate:**
Identify the two objects described in the prompt (e.g., "red cube" and "green sphere").
Understand the spatial relationship using 3D positioning:
"in front of" → Object A is closer to the viewer than Object B
"behind" or "hidden by" → Object A is farther away than Object B, regardless of whether it's partially visually obscured or not
Ignore visual occlusion — an object can still be considered "hidden by" another object if it is clearly located behind it in 3D space, even if visible.

**Scoring Criteria (0–2):**
2 = Correct - The relative 3D positions match the prompt clearly.
1 = Partially Correct - The 3D relationship is somewhat consistent but ambiguous or hard to judge.
0 = Incorrect - The spatial relationship is clearly wrong or objects are missing.

**Additional Notes:**
Focus on relative position in 3D space, not on visibility or occlusion.
If you're unsure about depth ordering, choose 1 and leave a short comment.
Ignore visual rendering quality, shadows, or object realism.

**Output format:** First think about it in steps following the above instructions, then give a one line answer as follows:

"Score = 2"

---

Figure 24: LLM instructions for evaluating 3D-spatial relations.

# G LLM USAGE

We use LLMs for three main purposes:

- **Methodology.** We use LLMs to automatically generate 2.5D image layouts from text.
- **Evaluation.** We use LLMs to automatically evaluate images for one of our metrics.
- **Writing.** We used LLMs to check grammar and refine phrasing during writing of this paper.

