# OpenReview forum: "ComposeAnything: Composite Object Priors for Text-to-Image Diffusion Models"
_ICLR.cc/2026/Conference — ICLR 2026 Conference Withdrawn Submission_

### Official Review · Reviewer_Rwmd · 2025-10-21

**Soundness:** 2
**Presentation:** 3
**Contribution:** 2
**Rating:** 4
**Confidence:** 4

**Summary:**

The paper tackles an important challenge in text-to-image generative models, achieving correct compositionality and precise text-image alignment. The authors propose a framework that employs a large language model (LLM) as a high-level planner to first produce a scene layout, accompanied by localized captions for each component and corresponding depth estimations. These structured elements are then used to generate individual image parts separately, which are subsequently composed into a coherent whole. The resulting composite image serves as a prior to guide the diffusion process, enhancing the compositional consistency.

**Strengths:**

- The paper clearly presents and thoroughly explains the proposed method, leaving no ambiguity.
- Ablation studies are conducted to evaluate the individual and combined effects of prior reinforcement and spatial control used in their method.
- The proposed approach demonstrates improvements over baseline models across evaluation metrics.
- The paper evaluates performance not only on benchmarks such as T2I-CompBench and NSR-1K, but also through human evaluations.

**Weaknesses:**

- The method lacks strong novelty, as it largely combines previously explored ideas, such as using an LLM as a planner, generating individual objects as conditional inputs, and employing object-prior reinforcement and spatially controlled denoising, all of which have been discussed in earlier works.
- The proposed framework has limited practical applicability. It requires multiple computationally intensive steps, including invoking an LLM, generating separate images for each component, performing segmentation, and adding extra computation during the main diffusion process, making it impractical for real-world or large-scale deployment.
- The evaluation is not thorough, as the paper only covers selected subsets of T2I-CompBench, omitting other aspects such as attribute binding (i.e., color, texture, shape categories of T2I-CompBench).
- The authors argue that fine-tuned models (e.g., FlowGRPO) perform worse in terms of image quality, yet this claim is supported only by qualitative evidence without any quantitative evaluation to substantiate it.
- Although the paper mentions generating and segmenting all objects in parallel using multiple GPUs to reduce computation time, this setup makes the reported runtime comparisons unfair. A fair evaluation should consider inference time on a single GPU to accurately reflect the method’s efficiency.

**Questions:**

- I noticed that the RPG results reported in the paper differ from those in the original RPG publication. Could you clarify whether these results were re-evaluated to verify their accuracy or directly taken from the original paper?

---

### Official Review · Reviewer_gqK8 · 2025-10-29

**Soundness:** 3
**Presentation:** 3
**Contribution:** 2
**Rating:** 4
**Confidence:** 3

**Summary:**

This paper proposed ComposeAnything, a training-free framework to enhance compositional image generation. It first uses LLM to plan for object boxes and depths, then generates corresponding objects as priors. A novel prior-guided diffusion is then leveraged to generate the final image. Extensive experiments demonstrate that ComposeAnything outperforms state-of-the-art methods across multiple compositional benchmarks.

**Strengths:**

1. The proposed method is conceptually intuitive and easy to understand. The idea of introducing structured object priors to guide diffusion generation is simple yet effective.
2. ComposeAnything achieves consistent quantitative and qualitative improvements over prior state-of-the-art methods across various compositional benchmarks.

**Weaknesses:**

1. The method shows limited novelty. The LLM-based layout planning and spatial-controlled attention are largely borrowed from RPG, and such mechanisms have already been widely used in recent compositional generation approaches.
2. The idea of replacing noise initialization with a composite object prior, though intuitive and easy to follow, is conceptually similar to FreeCompose[1], which also optimizes a composed prior image within the diffusion process.
3. The additional step of generating per-object priors introduces considerable computational overhead and latency.



[1] Chen, Zhekai, et al. "Freecompose: Generic zero-shot image composition with diffusion prior." European Conference on Computer Vision. Cham: Springer Nature Switzerland, 2024.

**Questions:**

The paper claims that “RPG uses region crops instead of object layout, so its LLM planning is less efficient than ours.”
Could the authors clarify why this is the case?
From the description, both methods seem to require the LLM to output multiple object positions (and ComposeAnything even adds depth ordering). So why is RPG less efficient?

---

### Official Review · Reviewer_4HMv · 2025-11-01

**Soundness:** 3
**Presentation:** 3
**Contribution:** 2
**Rating:** 4
**Confidence:** 3

**Summary:**

The paper proposes Plan-and-Paint, a framework combining two reasoning paradigms in T2I task for better image generation: (1) semantic-level planning via an Adaptive Length Prediction for CoT (ALP-CoT) that expands the prompt only as needed, and (2) noise-level reasoning trained with GRPO so the initial noise prior aligns with the semantic plan. A multi-reward ensemble (HPSv2 for preference, GroundingDINO for existence/count/relations, and a VQA model for attribute/theme) is used as the reward signal. On GenEval and WISE, the method reports SOTA performance and the ablations indicate ALP-CoT is beneficial: fixed-length CoT either over- or under-elaborates and hurts spatial tasks.

**Strengths:**

+Modular two-level design and targeted rewards. The pipeline (Fig. 2) and reward breakdown (Fig. 3) are modular and reasonable; the ensemble explicitly covers aesthetic, composition (existence/count/relations), and attribute/theme fidelity

+Strong results on GenEval/WISE versus both open and closed baselines; GenEval tops three of six subtasks, with large Position/Attr-Binding gains.

+Ablation studies validated the modular design, for example, Length sweeps (L=30…2048) show how fixed-length CoT fails on spatial reasoning.

**Weaknesses:**

-My first concern is that the novelty is largely integrative: ALP-CoT extends prior semantic CoT approaches (e.g., T2I-R1) with adaptive length; noise-level reasoning follows the idea of optimizing the initial noise prior (e.g., NoiseAR); GRPO is an existing RL algorithm. The contribution is a solid composition of known pieces rather than a new core algorithm.

-The ensemble is well-motivated, but the paper does not quantify robustness to reward drift/hallucinations or report sensitivity to reward weights, or is the method robust to the change of the reward weights?

-Qualitative prompts are mostly short, single-sentence directives: Examples across Figs. 4, 6, 7 use one-liners like “A photo of a bird below a skateboard,” “A photo of four computer keyboards,” etc. The experiment lacks long, fine-grained, multi-clause prompts (styles, attributes, scene context), which is important where the user needs fine-grained control for the generation.

-Compute/efficiency trade-offs is unknown (and might be higher than the baselines). GRPO with N=8, plus multi-expert scoring, implies non-trivial training cost; the paper doesn’t report inference-time overhead relative to the base generator (Qwen-Image)

**Questions:**

-For the reward weights, are they carefully tuned for different evaluation metrics/tasks?

-From the ablation studies, although there are ablations for with/without ALP-CoT and with/without noise-reasoning (NR), a dedicated fine-grained/long-prompt could be helpful to manifest the controllability of the generation.

---

### Official Review · Reviewer_XjjY · 2025-11-03

**Soundness:** 2
**Presentation:** 2
**Contribution:** 2
**Rating:** 4
**Confidence:** 4

**Summary:**

This paper proposes ComposeAnything, a training-free framework for compositional text-to-image generation. The method replaces random Gaussian noise with composite object priors — structured initializations built by combining objects generated from pretrained diffusion models, arranged using LLM-derived 2.5D layouts. During sampling, two mechanisms are introduced: (1) object-prior reinforcement and (2) spatial-controlled attention, both intended to better preserve compositional structure. Experiments on T2I-CompBench and NSR-1K show modest quantitative improvements and visually more coherent multi-object results compared to prior inference-only approaches.

**Strengths:**

Simple, practical design: The approach can be easily applied to existing diffusion backbones (e.g., SDXL, SD3, FLUX) without additional training.

Reasonable empirical validation: The paper provides quantitative and qualitative comparisons, as well as ablation studies for the two proposed modules.

Directionally relevant: The work contributes to a growing trend of inference-time structure control — integrating symbolic reasoning (via LLMs) with generative diffusion processes.

Readable and reproducible: The implementation details are clear and the modular design is straightforward for replication.

**Weaknesses:**

1. Limited originality — essentially RealCompo + RPG + InitNO.

The main components of ComposeAnything are all well-established:

LLM-based prompt decomposition and regional layout reasoning come directly from RPG (Yang et al., 2024), which already performs prompt parsing and region-wise diffusion using MLLMs.

Composite priors are conceptually equivalent to the layout-conditioned inputs used in RealCompo (Zhang et al., 2024), which integrates spatial-aware priors during inference for improved compositional control.

Noise initialization as an optimization variable mirrors InitNO (Guo et al., CVPR 2024), which formalized structured or optimized noise for compositional alignment.

ComposeAnything combines these three ingredients (layout → structured noise → spatial attention) in a straightforward way without introducing a new modeling mechanism or theoretical insight. The technical novelty is minimal.

2. Dependence on external systems.
The system’s success hinges heavily on GPT-4.1 for layout reasoning.

3. Lack of mechanistic understanding.
The paper doesn’t analyze why composite priors improve generation — whether they guide global attention, stabilize denoising, or simply bias the output distribution. Without such analysis, it’s unclear what the method contributes beyond empirical heuristics.

4. Evaluation is adequate but not conclusive.
Improvements on internal benchmarks (T2I-CompBench, NSR-1K) are moderate and not validated with broader or standardized compositional datasets. Missing baselines such as LayoutDiffusion or ControlNet limit the scope of the claimed “state-of-the-art” status.

**Questions:**

How does the method differ technically and conceptually from partial-denoising approaches like RPG?
To what extent do improvements depend on GPT-4.1’s layout accuracy?
Can the model handle complex overlapping objects or scenes with more than 5–10 regions?
Does the “composite prior” generalize to other modalities (e.g., depth, segmentation) or only visual layout?

---

### Note · Authors · 2025-11-14

I have read and agree with the venue's withdrawal policy on behalf of myself and my co-authors.